# Identity-Based Proxy Re-Encryption Scheme Using Fog Computing and Anonymous Key Generation

**DOI:** 10.3390/s23052706

**Published:** 2023-03-01

**Authors:** Han-Yu Lin, Tung-Tso Tsai, Pei-Yih Ting, Yan-Rong Fan

**Affiliations:** Department of Computer Science and Engineering, National Taiwan Ocean University, Keelung 202, Taiwan

**Keywords:** identity based, proxy re-encryption, fog computing, anonymous, cloud

## Abstract

In the fog computing architecture, a fog is a node closer to clients and responsible for responding to users’ requests as well as forwarding messages to clouds. In some medical applications such as the remote healthcare, a sensor of patients will first send encrypted data of sensed information to a nearby fog such that the fog acting as a re-encryption proxy could generate a re-encrypted ciphertext designated for requested data users in the cloud. Specifically, a data user can request access to cloud ciphertexts by sending a query to the fog node that will forward this query to the corresponding data owner who preserves the right to grant or deny the permission to access his/her data. When the access request is granted, the fog node will obtain a unique re-encryption key for carrying out the re-encryption process. Although some previous concepts have been proposed to fulfill these application requirements, they either have known security flaws or incur higher computational complexity. In this work, we present an identity-based proxy re-encryption scheme on the basis of the fog computing architecture. Our identity-based mechanism uses public channels for key distribution and avoids the troublesome problem of key escrow. We also formally prove that the proposed protocol is secure in the IND-PrID-CPA notion. Furthermore, we show that our work exhibits better performance in terms of computational complexity.

## 1. Introduction

Cloud computing was first introduced by Chellappa [1] who hoped to store data in remote servers via networks. In 2006, Amazon brought in the so-called Amazon Web Services (AWS) to seek more business opportunities. Since then, such services have been widely developed and spread out. Many companies like Microsoft and Google are gradually competing for the cloud computing market due to its advantages in reducing costs and improving productivity. Cloud services have also become an inevitable part of our daily life. When using cloud services, we have to pay attention to the safeguard of Internet security [2] by employing all kinds of cryptographic techniques [3,4,5,6].

Clouds are not only for storage; the deployment model influences managing and owning the cloud, and the location and users of the cloud. According to the location of stored data and how the technologies are deployed and consumed, we can classify cloud service models into three kinds, as follows:(i)**Public Cloud:** It is usually constructed by third-party cloud service companies (such as Google (Mountain View, CA, USA), Azure (Redmond, WA, USA), etc.). Users can purchase storage space from service providers and the latter will be responsible for system maintenance, which helps with reducing unnecessary user costs. Yet, the security is low owing to uncontrollable cloud environments.(ii)**Private Cloud:** It is constructed by the individual company and hence has high security and privacy. However, it requires self-maintenance and the cost is relatively high.(iii)**Hybrid Cloud:** It combines the advantages of public and private clouds and can separately store data by its confidentiality. Nevertheless, it is also relatively difficult to manage and maintain.

According to the definition given in NIST SP 800-145, the cloud service model is also categorized into the following three kinds:(i)**Software as a Service (SaaS):** It is the most common model in which users can utilize all kinds of interfaces (including web-based or program-based) to acquire resources and web services [7] such as stream media platforms running on cloud infrastructure. The advantage of this model is that users do not have to be responsible for controlling or maintaining the cloud infrastructure, such as communication networks [8], operating systems, storage, and applications.(ii)**Platform as a Service (PaaS):** In this model, the cloud service provider is responsible for providing application development platforms such as storage capacity, computing resources, programming languages, libraries and related development tools, etc. Users can utilize these tools to deploy consumer-created application programs on the cloud infrastructure and they do not have to control or maintain the cloud infrastructure.(iii)**Infrastructure as a Service (IaaS):** The cloud service provider supplies users with all kinds of storage, computing, and network resources, and users can utilize these infrastructures to deploy their own platforms and application programs. The advantage of this model is that users do not have to control or maintain the cloud infrastructure, but have the control over their deployed applications, storage, and operating systems.

Fog computing in IoT environments [9,10,11] is an extension of cloud computing and was first addressed by the research of Stolfo et al. [12], who attempt to protect the cloud data security [13] with the assistance of fog computing. Bonomi et al. [14] viewed the fog as a cloud closer to the ground and users. The architecture of fog computing can also benefit by sharing cloud data. A proxy re-encryption (PRE) scheme [15] is a commonly adopted ciphertext sharing protocol in which a ciphertext intended for Alice can be re-encrypted into another ciphertext designated for Bob by a proxy. When we combine the PRE scheme with fog computing, a fog would act as the proxy to carry out the cloud ciphertext transformation process, so as to reduce the network transmission latency. However, the privacy of cloud ciphertexts in the transformation must be further assured and the fog node (proxy) should learn nothing about the ciphertext.

## 2. Related Work

In 2010, Luo et al. [16] developed a ciphertext-policy attribute-based PRE scheme using AND-Gates policy. In particular, their policy supports multi-value attributes, negative attributes, and wildcards. They also showed that their mechanism fulfills the property of unidirectionality, non-interactivity, and multi-use. Moreover, in their scheme, the encryptor can decide if the ciphertext can be re-encrypted.

Considering the property of keyword search, in 2012, Fang et al. [15] proposed a chosen-ciphertext secure-anonymous-conditional PRE with keyword search. That is, they provided the PRE mechanism with the property of keyword search. Additionally, they gave the CCA security definition of conditional PRE schemes and showed that their protocol satisfies such a definition. Wang et al. [17] further introduced a constrained PRE scheme with a conjunctive keyword search. Specifically, their mechanism is both single-hop and unidirectional.

In 2013, Liang et al. [18] addressed a ciphertext-policy attribute-based PRE with chosen-ciphertext security assuming the hardness of the decisional *q*-parallel bilinear Diffie–Hellman exponent problem. In this mechanism, a ciphertext with respect to a given access policy is able to be re-encrypted into one in relation to another access policy. Their scheme is suitable for any monotonic access structure. They proved the security of their scheme in the random oracle model. Considering the queries from intra-domain and inter-domain in a cloud computing scenario, Han et al. [19] proposed an identity-based PRE scheme. Their scheme is secure against collusion attacks and the access permission could be made by the data owner, rather than by the central authority. However, the computational complexity of their scheme is high.

In 2014, Liang et al. [20] presented an adaptively CCA-secure ciphertext-policy attribute-based PRE for cloud data sharing. Their work integrated the dual system encryption technology with selective proof technique to achieve the adaptive CCA security in the standard model. Additionally, their work supports any monotonic access structures.

To improve the security of sharing data using QR codes, Akhil et al. [21] combined the PRE mechanism with QR code applications. Using QR codes to share data among different users is a commonly utilized approach. However, it is easily altered during transmission, since the format of QR codes is only readable by machines.

For improving the security of cloud storage, in 2018, Zeng and Choo [22] introduced a new conditional PRE scheme that is also known as the sender-specified PRE, i.e., SS-PRE, since their scheme only allows the proxy to transform the ciphertexts of the designated sender to the delegatee. They also present the formal definition of their SS-PRE scheme and prove its security in the standard model.

In order to share data securely in the cloud, in 2020, Zhang et al. [23] proposed an identity-based data storage scheme combining the architecture of fog computing. In their scheme, the fog node sends the request of the data user to both the cloud and the data owner. If the requested data user is non-revoked and has access privilege, the data owner will delegate the fog node to perform the cloud ciphertext re-encryption process. The fog node then forwards the re-encrypted ciphertext to the data user for decryption.

Considering the security of message transmission in a group, in 2021, Xiong et al. [24] presented a so-called puncturable PRE scheme on the identity-based cryptosystem. In particular, there is a message server that carries out the ciphertext re-encryption for all users and thus the computational efforts of the user sides are released. Yet, the message server plays a crucial role in the system performance and might become an obvious attacking target.

In 2022, Lin et al. [25] improved Zhang et al.’s scheme [23] by eliminating some security flaws. They also showed that their enhanced scheme maintains the properties to revoke invalid users and generate private keys anonymously. Although their work has improved security, which is provably secure in the random oracle model, the computational complexity is still high. Motivated by this challenging problem, we propose a more efficient PRE scheme based on the study of Lin et al. [25]. Up to the present, there have been several PRE mechanisms [26,27,28,29,30,31,32] proposed for different applications. However, only a few works [19,23,25] take the issue of cloud computing or fog computing scenarios into consideration. We compare the proposed mechanism with these schemes and show the computational advantage of ours in a later section.

The main contribution of this study is to propose an identity-based PRE scheme for the fog computing scenario and using the technique of anonymous key generation. In the proposed system, we use public channels for key distribution and avoid the troublesome problem of key escrow. In addition, the decision of access privilege for cloud ciphertexts is controlled by the data owner, rather than by the central authority. Moreover, we demonstrate that the proposed protocol is not only IND-PrID-CPA secure, but also has lower computational costs.

## 3. Preliminaries

Let the symbols (***G***_1_, ***G***_2_) be two multiplicative groups and *p* is a prime order of both groups. We express *e*: ***G***_1_ × ***G***_1_ → ***G***_2_ as a symmetric bilinear pairing. The properties of *e* are listed as follows:***(i)*** ***Bilinearity:***  

Given a group element *g* in ***G***_1_ and two integers *a*, *b* in *Z_p_*, we have *e*(*g^a^*, *g^b^*) = *e*(*g*, *g*)*^ab^*.
***(ii)*** ***Non-degeneracy:*** 

There are group elements *A*, *B* in ***G***_1_ such that *e*(*A*, *B*) ≠ 1.
***(iii)*** ***Computability:*** 

Given two group elements *A*, *B* in ***G***_1_, the value *e*(*A*, *B*) can be efficiently computed.

### Decisional Bilinear Diffie–Hellman (DBDH) Problem and Assumption

Let the problem instance be (*g*, *g^x^*, *g^y^*, *g^z^*, *e*(*g*, *g*)*^xyz^*, *C*) in which (*g*, *g^x^*, *g^y^*, *g^z^*) are elements in ***G***_1_ while (*e*(*g*, *g*)*^xyz^*, *C*) are elements in ***G***_2_; the DBDH problem has to decide if the equality *e*(*g*, *g*)*^xyz^* = *C* holds or not. Its assumption asserts that the chance of any adversary running in polynomial-time to solve the DBDH problem is insignificant.

## 4. Proposed IB-PRE-FCAK Scheme

Before introducing the proposed identity-based proxy re-encryption scheme using fog computing and anonymous key generation (short for IB-PRE-FCAK), we first address the system model and composed algorithms.

### 4.1. System Model

We illustrate the system model for our proposed IB-PRE-FCAK scheme in Figure 1, and it is mainly composed of three levels. Among the hierarchy, the top level is a cloud server that can be viewed as a data repository center storing ciphertexts. The middle level is a collection of fog nodes that process requests from users and transmit ciphertexts to the cloud server. The third level consists of the data owner and the data user. The former generates ciphertexts to be uploaded to the cloud server while the latter can request access to cloud ciphertexts. Note that both the ciphertext uploading and downloading processes are assisted by the fog nodes. In particular, the data owner can authorize the fog node to perform the ciphertext re-encryption procedure for sharing cloud ciphertexts with other data users. Moreover, there is also a private key generation center (PKG) responsible for issuing private keys to all users.

### 4.2. Algorithms

The IB-PRE-FCAK scheme can be divided into several subroutines, i.e., Setup, KeyExtract, Enc, Tkgen, RKgen, Re-Enc, and Dec. We define the parameters and corresponding outputs of these subroutines as follows:**–** **Setup**(1*^l^*)**:** This subroutine utilizes the value *l* as a security parameter and returns the system public information Φ along with the master secret key *Msk*.**–** **KeyExtract**(Φ, *Msk*, *ID*)**:** This subroutine utilizes three input parameters (Φ, *Msk*, *ID*) where *ID* is a user identity, and performs an interactive process to return the private key *K_ID_* associated with *ID*.**–** **Enc**(Φ, *ID*, *m*, *SK*)**:** This subroutine utilizes four input parameters (Φ, *ID*, *m*, *SK*) where (*m*, *SK*) separately represents a plaintext and a symmetric encryption key. It returns a ciphertext *C* of the plaintext *m* under the key *SK*.**–** **Tkgen**(Φ, *ID_u_*, *K_ID__u_*, *C_ind_*)**:** This subroutine utilizes four input parameters (Φ, *ID_u_*, *K_ID__u_*, *C_ind_*) where *C_ind_* is the name of data category, and then returns a token *T_u,ind_*.**–** **RKgen**(Φ, *ID_u_*, *K_ID__o_*, *T_u,ind_*)**:** This subroutine utilizes five input parameters (Φ, *ID_u_*, *K_ID__o_*, *T_u,ind_*) where *ID_o_* is the data owner. It returns either the symbol of error ⊥ or a corresponding key *RK_o,u,ind_* for re-encryption.**–** **Re-Enc**(Φ, *ID_u_*, *C*, *RK_o,u,ind_*)**:** This subroutine utilizes four input parameters (Φ, *ID_u_*, *C*, *RK_o,u,ind_*) and then returns a re-encrypted ciphertext *C′*.**–** **Dec**(Φ, *K_ID_*, *C**)**:** This subroutine utilizes three input parameters (Φ, *K_ID_*, *C**) where *C** could be *C* or *C′*, and then returns a plaintext *m*.

### 4.3. Construction

We introduce a concrete construction based on the previously defined subroutines. First, some utilized symbols are defined as Table 1:
**–** **Setup:** Taking the value *l* as a security value, the PKG chooses ***G***_1_ and ***G***_2_ groups of prime order *p* and both are multiplicative. Let the symbol *g* denote a generator in group ***G***_1_ and the notation *e* be a bilinear map written as *e*: ***G***_1_ × ***G***_1_ → ***G***_2_. *Msk* determined by the PKG is a random value *s* ∈ *Z_p_**, and its corresponding master public key (*Mpk*) is calculated as *Q* = *g^s^*. There is also a user revocation list, i.e., *RL*, maintained by the PKG. Whenever *ID_i_* has to be revoked, the PKG renews *RL* as *RL*′ = *RL* ∪ {*ID_i_*}. Three collision-resistant hash functions are defined as follows:

*h_i_*: {0, 1}* → ***G***_1_, for *i* = 1 and 2

*h*_3_: ***G***_2_ → ***G***_1_

Except for *Msk*, all the other parameters could be viewed as the system public information Φ.
**–** **KeyExtract:** For obtaining his/her private key, a user *ID_i_* randomly chooses integers *d_i_*, *k_i_*∈ *Z_p_** and computes

*D_i_* = *g^di^*, (1)

*H*′*_i_* = *D_i_*·*h*_1_(*ID_i_* ‖ *k_i_*) (2)

The values (*ID_i_*, *H*′*_i_*) are delivered to the PKG. After receiving it, the PKG derives
*K*′*_i_* = (*H*′*_i_*·*h*_2_(*ID_i_* ‖ *ID_PKG_*))^*s*^,(3)
and sends *K*′*_i_* back to *ID_i_*. Consequently, *ID_i_* is able to calculate his private key as
*K_i_* = *K′_i_*/(*Q*)*^di^* = (*h*_1_*(ID_i_*‖ *k_i_*)*·h*_2_(*ID_i_*‖ *ID_PKG_*))*^s^*(4)

With the following equality, the correctness of *K_i_* can be easily verified.
*e*(*K_i_*, *g*) = *e*(*h*_1_(*ID_i_* ‖ *k_i_*)·*h*_2_(*ID_i_* ‖ *ID_PKG_*), *Q*)(5)
**–** **Enc:** Let *m* = (*m*_1_, *m*_2_, …, *m_n_*) be a plaintext to be encrypted and *SK* ∈ ***G***_2_ a chosen symmetric key. A data owner *ID_o_* then selects an integer *z* ∈ *_R_ Z_p_** to calculate

*r*_1_ = *SK* · *e*(*Q*, (*h*_1_(*ID_o_* ‖ *k_o_*)·*h*_2_(*ID_o_* ‖ *ID_PKG_*))*^z^*),(6)*r*_2_ = *g^z^*, (7)*r*_3_ = (*E*(*SK*, *m*_1_), *E*(*SK*, *m*_2_), …, *E*(*SK*, *m_n_*))(8)
where the notation *E*(·) denotes the symmetric encryption function.

Here, the ciphertext *C* is composed of {*r*_1_, *r*_2_, *r*_3_}. Next, the data owner sends (*ID_o_*, *C*) along with the data category name *C_ind_* to the adjacent fog. It stores (*ID_o_*, *C_ind_*, *r*_1_, *r*_2_) in the local repository and further transmit (*ID_o_*, *C_ind_*, *r*_3_) to the cloud server.
**–** **Tkgen:** For accessing the cloud data with respect to the data category name *C_ind_*, a data user *ID_u_* randomly selects an integer *r* ∈ *_R_ Z_p_** to compute

*R* = *g^r^*,(9)
and delivers his request (*ID_u_*, *C_ind_*, *R*) to the adjacent fog. It then searches for the record (*ID_o_*, *C_ind_*, *r*_1_, *r*_2_) from the local repository and further forwards the token *T_u,ind_* = (*ID_u_*, *R*) to the associated data owner *ID_o_*.
**–** **RKgen:** Upon obtaining the token *T_u,ind_* = (*ID_u_*, *R*), the data owner *ID_o_* asks the PKG to check if the maintained user revocation list *RL* contains *ID_u_*. If it does, an error symbol ⊥ is sent to the requested data user *ID_u_* via the assistance of the fog. Or else, *ID_o_* randomly selects two random numbers *t*, *y* ∈ *Z_p_** and computes

*w*_1_ = *Q^t^*(10)

(11)$$w_{2}=\frac{K_{o}w_{1}}{h_{3}\left(e\left(h_{2}(ID_{u}||ID_{PKG})R^{y},\;\;Q\right)\right)}\;$$*w*_3_ = *e*(*g^y^*, *Q*)(12)
Next, *ID_o_* sends the re-encryption key *RK_o,u,ind_* = (*w*_1_, *w*_2_, *w*_3_) to the fog node.


**–** **Re-Enc:** Upon receiving *RK_o,u,ind_*, the fog re-encrypts the original ciphertext *C* as *C*′ by setting


*r*_1_′ = *r*_1_·*e*(*w*_1_, *r*_2_), (13)

*r*_4_′ = *w*_2_, (14)

*r*_5_′ = *w*_3_(15)

At last, the re-encrypted ciphertext *C*′ consisting of (*r*_1_′, *r*_2_, *r*_3_, *r*_4_′, *r*_5_′) is sent back to *ID_u_*. Note that the partial ciphertext *r*_3_ can be retrieved from the cloud storage.
**–** **Dec:** In the case that the data owner *ID_o_* wants to access his/her original ciphertext *C* = (*r*_1_, *r*_2_, *r*_3_), he/she can derive the symmetric key by computing


(16)
$$SK=\frac{r_{1}}{e\left(r_{2},\;K_{o}\right)}$$


Then, the plaintext can be decrypted as
*m* = (*m*_1_, *m*_2_, …, *m_n_*) = (*D*(*SK*, *r*_3, 1_), *D*(*SK*, *r*_3, 2_), …, *D*(*SK*, *r*_3,_
*_n_*)).(17)
where *D*(·) is a symmetric decryption function.

The correctness of Equation (16) can be verified as follows. From the right side of Equation (16), it can be derived that
$$\begin{array}{cc} \frac{r_{1}}{e\left(r_{2},K_{o}\right)} & =\frac{SK\cdot e\left(Q,\;{\left(h_{1}(ID_{O}||k_{O})h_{2}(ID_{O}||ID_{PKG})\right)}^{z}\right)}{e\left(g^{z},{\left(h_{1}(ID_{O}||k_{O})h_{2}(ID_{O}||ID_{PKG})\right)}^{s}\right)}\\ & =\frac{SK\cdot e(g^{sz},\;(h_{1}(ID_{O}||k_{O})h_{2}(ID_{O}||ID_{PKG})))}{e\left(g^{z},{\left(h_{1}(ID_{O}||k_{O})h_{2}(ID_{O}||ID_{PKG})\right)}^{s}\right)}\\ & =SK \end{array}$$

Whenever a data user *ID_u_* obtains a re-encrypted ciphertext *C*′ that is composed of (*r*_1_′, *r*_2_, *r*_3_, *r*_4_′, *r*_5_′), he/she first utilizes his/her own private key *K_u_* to calculate
(18)$$X={r_{4}}^{\prime }\cdot h_{3}\left(\frac{{r^{\prime }}_{5}^{r}\cdot e\left(K_{u},\;g\right)}{e\left(h_{1}(ID_{u}||k_{u}),\;\;Q\right)}\right)$$
(19)$$SK=\frac{{r_{1}}^{\prime }}{e\left(X,\;\;r_{2}\right)}.\;$$

Consequently, the plaintext *m* can be decrypted with the symmetric key *SK* by Equation (17). We give the derivations of Equation (19) below. Our first step is to simplify Equation (18):$$\begin{array}{cc} X & ={r_{4}}^{\prime }\cdot h_{3}\left(\frac{{r^{\prime }}_{5}^{r}\cdot e\left(K_{u},\;g\right)}{e\left(h_{1}(ID_{u}||k_{u}),Q\right)}\right)\\ & =\frac{K_{o}w_{1}}{h_{3}\left(e\left(h_{2}(ID_{u}||ID_{PKG})R^{y},Q\right)\right)}h_{3}\left(\frac{e\left(g^{yr},\;Q\right)e\left(K_{u},g\right)}{e\left(h_{1}(ID_{u}||k_{u}),Q\right)}\right)\\ & =\frac{K_{o}w_{1}}{h_{3}\left(e\left(h_{2}(ID_{u}||ID_{PKG})R^{y},Q\right)\right)}h_{3}\left(\frac{e\left(g^{yr},\;Q\right)e\left(h_{1}(ID_{u}||k_{u})h_{2}(ID_{u}||ID_{PKG}),g^{s}\right)}{e\left(h_{1}(ID_{u}||k_{u}),Q\right)}\right)\\ & =\frac{K_{o}w_{1}}{h_{3}\left(e\left(h_{2}(ID_{u}||ID_{PKG})R^{y},Q\right)\right)}h_{3}\left(e\left(R^{y}h_{2}(ID_{u}||ID_{PKG}),\;Q\right)\right)\\ & =K_{o}w_{1} \end{array}$$

Next, we could rewrite Equation (19) as
$$\begin{array}{cc} \frac{{r_{1}}^{\prime }}{e\left(X,\;r_{2}\right)} & =\frac{r_{1}\cdot e\left(w_{1},\;r_{2}\right)}{e\left(K_{o}w_{1},r_{2}\right)}\\ & =\frac{SK\cdot e\left(Q,{\left(h_{1}(ID_{O}||k_{O})h_{2}(ID_{O}||ID_{PKG})\right)}^{z}\right)e\left(Q^{t},g^{z}\right)}{e\left({\left(h_{1}(ID_{O}||k_{O})h_{2}(ID_{O}||ID_{PKG})\right)}^{s}Q^{t},g^{z}\right)}\\ & =\frac{SK\cdot e\left({\left(h_{1}(ID_{O}||k_{O})h_{2}(ID_{O}||ID_{PKG})\right)}^{s}Q^{t},g^{z}\right)}{e\left({\left(h_{1}(ID_{O}||k_{O})h_{2}(ID_{O}||ID_{PKG})\right)}^{s}Q^{t},g^{z}\right)}\\ & =SK \end{array}$$

## 5. Formal Model and Security Proof

The fundamental security for any encryption scheme is confidentiality. There is also a well-defined security model for PRE schemes. We adopt this security model to demonstrate formally the security of the proposed IB-PRE-FCAK protocol. Namely, we initially review the security definition of IND-PrID-CPA, i.e., indistinguishability against adaptively chosen identity and chosen plaintext attacks. Then, we demonstrate that the proposed construction fulfills the secure notion of IND-PrID-CPA by utilizing the proof techniques of random oracle models.

The concept of this security proof is a technique of proof by contradiction. That is, we first assume that there is an adversary who is able to break the proposed scheme under the adaptively chosen identity and chosen plaintext attacks. Then, we can take the advantage of this adversary to break a well-known intractable cryptographic assumption with non-negligible advantage. Since there is no efficient polynomial-time algorithm that could solve any well-known cryptographic assumption, we conclude that our initial assumption is wrong, which also completes the whole security proof.

**(IND-PrID-CPA)** *In the following interactive game between a probabilistic adversary* A *and a challenger* B*, if the former does not have a non-negligible advantage to defeat the latter in polynomial-time, we say that an IB-PRE-FCAE scheme fulfills the security requirement of indistinguishability under the attacks of adaptively chosen identity and chosen-plaintext:*

**Setup:** At first, the challenger B invokes the Setup(1*^l^*) subroutine to obtain system public information Φ along with the master secret key *Msk*. The adversary A can only learn the public information Φ.

**Phase 1:** A is allowed to adaptively invoke the following queries:–*KeyExtract Queries:* A can query the private key for his chosen identity.–*RKgen Queries:* A can query the re-encryption key for his chosen (*ID_o_*, *ID_u_*, *C_ind_*) in which *ID_u_* has to be a non-revoked data user and *C_ind_* is the name of data category.

**Challenge:** A chooses the identity of *ID** as an object and prepares the plaintext *m** = (*m*_1_*, *m*_2_*, …, *m_n_**). Let (*SK*_0_, *SK*_1_) be symmetric keys with an identical length. Then, B flips a bit *bt* and then creates a challenge ciphertext *C** = (*r*_1_*, *r*_2_*, *r*_3_*) in relation to (*ID**, *m**, *SK_bt_*) for A.

**Phase 2:** Given the ciphertext *C**, the adversary A goes on to invoke previous queries based on the following limits:–The KeyExtract query with respect to *ID**, i.e., the target identity, is prohibited.–Any RKgen query for the identities (*ID**, *ID_u_*) or (*ID_o_*, *ID**) is prohibited.–A can invoke at most *q_ke_* KeyExtract and *q_rk_* RKgen queries.

**Guess:** After invoking enough queries, A returns a bit *bt*′. We say that A wins this game, provided that *bt*′ = *bt*. Therefore, we can express A’s advantage as *Adv*(A) = | Pr[*bt*′ = *bt*′] − 1/2 |.

Using the techniques of random oracle proof models, we prove that the proposed IB-PRE-FCAK scheme satisfies the security notion of IND-PrID-CPA as Theorem 1.

**Theorem** **1.** *Provided that**the DBDH assumption holds, the proposed IB-PRE-FCAK scheme satisfies the security requirement of indistinguishability under adaptively chosen identity and chosen plaintext attacks (IND-PrID-CPA). Specifically, an algorithm* B *breaking the DBDH problem with non-negligible advantage ε′ can be created by utilizing a probabilistic adversary* A *that is able to break the IND-PrID-CPA security of the proposed IB-PRE-FCAK scheme with non-negligible advantage ε in polynomial-time. To be precise, the non-negligible advantage ε′ can be expressed as*$$\varepsilon^{\prime }\;\geq \;\frac{\varepsilon }{e\left(q_{ke}+q_{rk}+1\right)}$$*where q**_ke_ and q**_rk_ are the maximum numbers of KeyExtract and RKgen queries, respectively.*

**Proof.** Given an instance (*g*, *g^a^*, *g^b^*, *g^c^*, *e*(*g*, *g*)*^abc^*, *F*) of DBDH, we build an algorithm B to judge if *e*(*g*, *g*)*^abc^* = *F* holds or not by taking the adversary A as subroutine. In the following interactions, B is responsible for responding to various queries submitted by A.**Setup:** At first, the challenger B invokes the Setup(1*^l^*) subroutine to obtain system public information Φ. Let (*h*_1_, *h*_2_) be random oracles and *h*_3_ a collision-resistant hash function. B further sets *Mpk* = *Q* = *g^a^*, i.e., *Msk* is implicitly specified as the value *a* unknown to B.**Phase 1:** A is allowed to adaptively invoke the following queries:–*h*_1_(*ID_i_* ‖ *k_i_*) *query:* In this query, B first searches the maintained *h*_1_-list for a matched record. Or else, he selects a bit *η* such that Pr[*η* = 1] = *τ*. The value *τ* would be derived subsequently. Whenever *η* = 0, B returns the value *J*_1_ = (*g^b^*)*^v^*^1^ in which *v*_1_ ∈ *Z_p_*^*^. Otherwise, *J*_1_ is computed as *g^v^*^1^. The maintained *h*_1_-list is also renewed by adding the record (*ID_i_*, *k_i_*, *η*, *v*_1_, *J*_1_).–*h*_2_(*ID_i_* ‖ *ID_PKG_*) *query:* In this query, B first searches the maintained *h*_2_-list for a matched record. Or else, he returns the value *J*_2_ = *g*^*v*2^ in which *v*_2_ ∈ *Z_p_*^*^. The maintained *h*_2_-list is also renewed by adding the record (*ID_i_*, *ID_PKG_*, *v*_2_, *J*_2_).–*KeyExtract query:* In response to the KeyExtract(*ID_i_*) query, B tries to determine the corresponding records (*ID_i_*, *k_i_*, *η*, *v*_1_, *J*_1_) and (*ID_i_*, *ID_PKG_*, *v*_2_, *J*_2_) in *h*_1_-list and *h*_2_-list, respectively. (If one datum exists, B could directly invoke the two queries to create records.) As long as *η* = 1, B aborts; or else, the return value is computed as *K_i_* = $Q^{\left(v_{1}\;+\;v_{2}\right)}$–*RKgen query:* In response to the RKgen(*ID_o_*, *ID_u_*, *C_ind_*) query in which *ID_u_* is a non-revoked user, B obtains the private key *K_ID__o_* by invoking the KeyExtract(*ID_o_*) query and checks the record (*ID_i_*, *k_i_*, *η*, *v*_1_, *J*_1_) kept in the *h*_1_-list. As long as *η* = 0, B aborts. Or else, B chooses random numbers *r*, *t*, *y* ∈ *Z_p_*^*^ and calculates *R* = *g^r^*, *w*_1_ = *Q^t^*, *w*_2_ = $\frac{K_{o}Q^{t}}{h_{3}\left(e\left(h_{2}(ID_{u}||ID_{PKG}))R^{y},\;Q\right)\right)}$, *w*_3_ = *e*(*g^y^*, *Q*). Thus, the returned re-encryption key *RK_o,u,ind_* is composed of (*w*_1_, *w*_2_, *w*_3_).**Challenge:** A chooses the identity of *ID** as an object and prepares the plaintext *m** = (*m*_1_*, *m*_2_*, …, *m_n_**). Let (*SK*_0_, *SK*_1_) be symmetric keys with an identical length. Then, B flips a bit *bt* and then creates a challenge ciphertext *C** = (*r*_1_*, *r*_2_*, *r*_3_*) in relation to (*ID**, *m**, *SK_bt_*) for A as follows:**Step 1** Suppose that the *h*_1_(*ID** ‖ *k**) query has been made. As long as *η** = 1, B directly aborts.**Step 2** Define *h*_2_(*ID** ‖ *ID_PKG_*) = *g^v^*^2^ in which *v*_2_ ∈ *Z_p_*^*^.**Step 3** Set the partial ciphertext *r*_2_* = *g^c^*.**Step 4** Determine the value *v*_1_ of the record (*ID**, *k**, *η**, *v*_1_, *J*_1_) in the *h*_1_-list and calculate

$${r_{1}}^{*}=SK_{bt}\cdot F^{v_{1}^{-1}}\cdot e\left(Q,\;{\left(g^{c}\right)}^{v_{2}}\right),\;$$

*r*_3_* = (*E*(*SK_bt_*, *m*_1_*), *E*(*SK_bt_*, *m*_2_*), …, *E*(*SK_bt_*, *m_n_**)).
Consequently, the returned challenge ciphertext is *C** = (*r*_1_***, *r*_2_***, *r*_3_***).**Phase 2:** Given the ciphertext *C**, the adversary A goes on to invoke queries based on the previous limitations.**Guess:** After invoking enough queries, A returns a bit *bt*′. In case that *bt*′ = *bt*, B directly returns 1, meaning that *F* = *e*(*g*, *g*)*^abc^*. Otherwise, the value 0 is outputted instead.**Analysis:** In these simulation processes, it can be observed that when *F* is equivalent to *e*(*g*, *g*)*^abc^*, the prepared challenge ciphertext *C** is a legal one. According to the initial assumption, A would have the non-negligible advantage to break the proposed IB-PRE-FCAK scheme provided that the simulated ciphertext *C** is valid. That is to say, we know that *Adv*(A) = | Pr[*bt*′ = *bt*] − 1/2 | ≥ *ε*. Yet, when *F* is not equivalent to *e*(*g*, *g*)*^abc^*, the advantage for A to output a correct bit *bt*′ is not superior, which implies that Pr[*bt*′ = *bt*] = 1/2. Therefore, the chance for B to solve the problem of DBDH could be written as
| Pr[(*g*, *g^a^*, *g^b^*, *g^c^*, *e*(*g*, *g*)*^abc^*) = 1] − Pr[(*g*, *g^a^*, *g^b^*, *g^c^*, *F*) = 1] |≥ | (1/2 + *ε*) − 1/2 |·Pr[Good]= *ε*·Pr[Good]
where Pr[Good] represents the probability event that B never aborts during the game interaction processes. To calculate Pr[Good], we further consider the following several cases:Pr[¬KeyExtract]: the likelihood that B never aborts in any KeyExtract query;Pr[¬RKgen]: the likelihood that B never aborts in any RKgen query;Pr[¬Challenge]: the likelihood that B never aborts in the challenge phase.In the first case of a KeyExtract query, B aborts as long as the bit *η* in the corresponding entry of the *h*_1_-list equals 0. Thus, we can learn that Pr[¬KeyExtract] ≤ *τ ^qke^*. Likewise, as for the second case of an RKgen query, we also know that B aborts on the condition that the bit *η* = 0, which indicates that Pr[¬RKgen] ≤ *τ ^qrk^*. In the third case that B might abort only when the bit *η** for the chosen identity *ID** is equivalent to 1. Therefore, we could derive that Pr[¬Challenge] ≤ (1 − *τ*). Putting all the three independent probability events together, we further obtain
Pr[Good] = Pr[¬KeyExtract]·Pr[¬RKgen]·Pr[¬Challenge]      ≤ (*τ*)*^qke^*(*τ*)*^qrk^*(1 − *τ*)      = (*τ*)*^qke^*
^+^
*^qrk^*(1 − *τ*).      = $\frac{1}{e\left(q_{pk}+q_{pr}+1\right)}$
To maximize the value of Pr[Good], we set *τ* to be $1-\frac{1}{q_{ke\;}+q_{rk}+1}$ such that Pr[Good] = $\frac{1}{e\left(q_{pk}+q_{pr}+1\right)}$ becomes the greatest value, where *e* denotes the base of natural logarithm. As a result, we claim that the constructed algorithm B has a non-negligible advantage *ε′* ≥ $\frac{\varepsilon }{e\left(q_{pk}+q_{pr}+1\right)}$ to break the DBDH problem. □

## 6. Efficiency and Comparison

We made some efficiency comparisons with related protocols [19,23,25] in terms of several time-consuming computations. For convenience, the simulation environments are listed in Table 2 and the compared computation is also converted into approximate running time in Table 3. The detailed evaluation results are summarized in Table 4 and Figure 2.

Although Han et al.’s scheme is cost-free in the Re-Enc phase, their scheme has the highest computation costs in the Setup, KeyExtract, Enc, TKgen, and RKgen phases. Zhang et al.’s scheme has the lowest computation costs in both the RKgen and the Dec by *ID_u_* phases. Lin et al.’s scheme incurs higher computation cost in the Dec by *ID_u_* phase. As a whole, the proposed protocol exhibits optimal computational costs in the Setup, KeyExtract, Enc, and Dec by *ID_o_* phases.

## 7. Conclusions

Fog-based applications have received much attention in recent years due to their advantages in fast response time and more bandwidth savings. A fog-enabled proxy re-encryption scheme allows a fog node to perform the ciphertext re-encryption process, so as to share cloud ciphertexts to desired data users. In this paper, we propose an identity-based proxy re-encryption scheme taking the advantage of fog computing. Specifically, the proposed scheme removes the necessity for a fully trusted system authority, as the private key of each user is not generated by the system authority solely. Therefore, it is unnecessary to establish a secure channel for distributing private keys in the proposed scheme. The access privilege of cloud ciphertexts can be determined independently by the data owner. As for security, we adopt the security notion of IND-PrID-CPA to formally prove that the proposed mechanism is able to withstand the adaptive adversary in random oracle models. In the performance analyses, we also demonstrate that our work is efficient in the processes of Setup, KeyExtract, Enc, and Dec, when compared with related protocols.

## Figures and Tables

**Figure 1 sensors-23-02706-f001:**
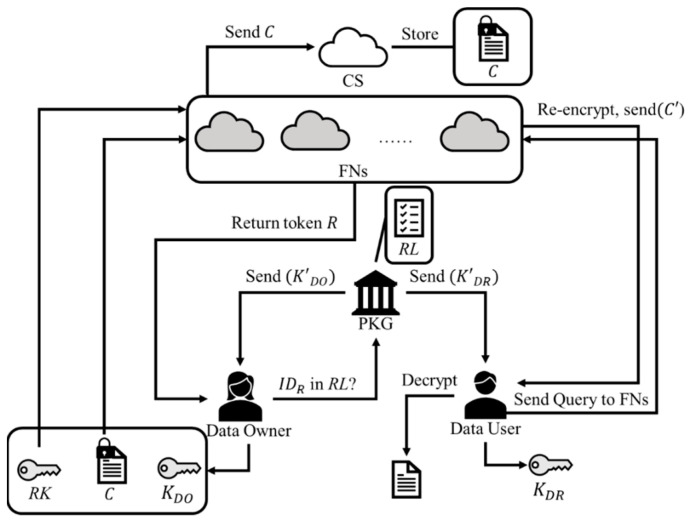
System model of the IB-PRE-FCAK scheme.

**Figure 2 sensors-23-02706-f002:**
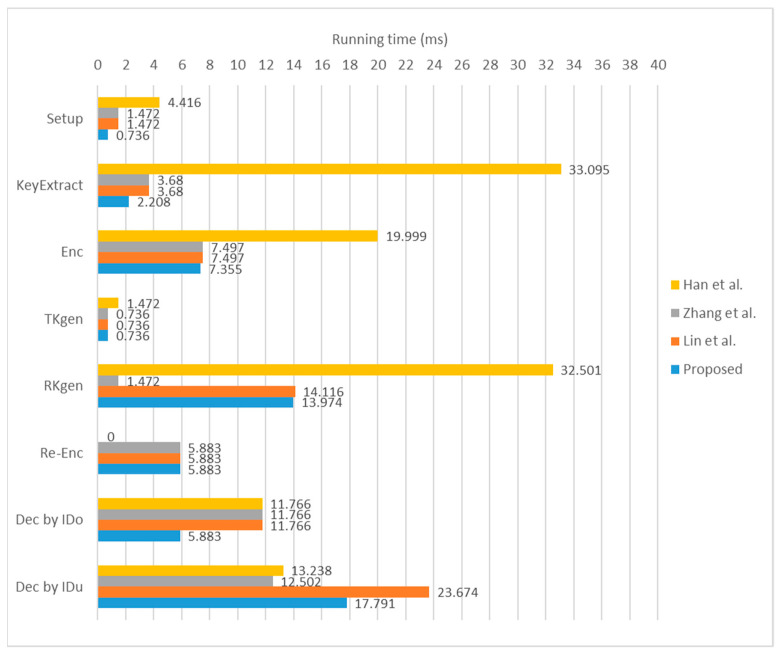
Comparison of approximate running time [11,13,19].

**Table 1 sensors-23-02706-t001:** Symbol notations.

Notation	Description
*l*	security value
***G***_1_, ***G***_2_	groups of prime order *p*
*g*	a generator of ***G***_1_
*e*	a bilinear map satisfying *e*: ***G***_1_ × ***G***_1_ → ***G***_2_
*s*	master secret key
*Q*	master public key satisfying *Q* = *g^s^*
*RL*	revocation list
*h*_1_, *h*_2_, *h*_3_	collision-resistant hash functions
Φ	system public information
*SK*	symmetric key
*E*(·)/*D*(·)	symmetric encryption/decryption function
{*r*_1_, *r*_2_, *r*_3_}	ciphertext
*C_ind_*	data category name
(*w*_1_, *w*_2_, *w*_3_)	re-encryption key
(*r*_1_′, *r*_2_, *r*_3_, *r*_4_′, *r*_5_′)	re-encrypted ciphertext

**Table 2 sensors-23-02706-t002:** Simulation environments.

Item	Environment
Processor	Intel Core 2 Duo @ 2.1 Ghz
Memory size	2 GB
Operating system	Linux Ubuntu version 9.1
Software	PBC library [33]

**Table 3 sensors-23-02706-t003:** Computation and approximate running time.

	**Item**	**Notation**	**Running Time**
**Computation**			
Bilinear pairing		*C* _0_	5.883 ms
Exponentiation over *G*_1_		*C* _1_	0.736 ms
Exponentiation over *G*_2_		*C* _2_	0.142 ms

**Table 4 sensors-23-02706-t004:** Evaluation of computational cost.

	Scheme	Han et al. [19]	Zhang et al. [11]	Lin et al. [13]	Proposed
Phase					
Setup cost		6*C*_1_	2*C*_1_	2*C*_1_	*C* _1_
KeyExtract cost		5*C*_0_ + 5*C*_1_	5*C*_1_	5*C*_1_	3*C*_1_
Enc cost		3*C*_0_ + 3*C*_1_ + *C*_2_	*C*_0_ + 2*C*_1_ + *C*_2_	*C*_0_ + 2*C*_1_ + *C*_2_	*C*_0_ + 2*C*_1_
Tkgen cost		2*C*_1_	*C* _1_	*C* _1_	*C* _1_
RKgen cost		5*C*_0_ + 4*C*_1_ + *C*_2_	2*C*_1_	2*C*_0_ + 3*C*_1_ + *C*_2_	2*C*_0_ + 3*C*_1_
Re-Enc cost		0	*C* _0_	*C* _0_	*C* _0_
Dec cost by *ID_o_*		2*C*_0_	2*C*_0_	2*C*_0_	*C* _0_
Dec cost by *ID_u_*		2*C*_0_ + 2*C*_1_	2*C*_0_ + *C*_1_	4*C*_0_ + *C*_2_	3*C*_0_ + *C*_2_

## Data Availability

Data is contained within the article.

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
