# Peer review of "Identity-Based Proxy Re-Encryption Scheme Using Fog Computing and Anonymous Key Generation"

_sensors, 2023, doi:10.3390/s23052706_

Round 1
Reviewer 1 Report
Overall, this is a clear, concise, and well-written manuscript. The introduction is relevant. Sufficient information about the previous study findings is presented for readers to follow the present study rationale and procedures. The methods are generally appropriate on the Title “Identity-based Proxy Re-encryption Scheme Using Fog Compu- 2 ting and Anonymous Key Generation”
clarification of a few details needed. Specific comments follow.
1. More description is required how the proposed protocol is secure in the IND-PrID-CPA notion.
2. More precise formulation needed how proposed mechanism is able to withstand the adaptive adversary in the random oracle models.
3. In point 5. Efficiency and Comparison The detailed of the simulation environment is required.
Author Response
The authors would like to thank reviewers for their valuable suggestions that result in the improvement of correctness and readability of this manuscript. Revisions with respect to each comment are listed below:
Comments of Reviewer 1:
Overall, this is a clear, concise, and well-written manuscript. The introduction is relevant. Sufficient information about the previous study findings is presented for readers to follow the present study rationale and procedures. The methods are generally appropriate on the Title “Identity-based Proxy Re-encryption Scheme Using Fog Computing and Anonymous Key Generation.”
Clarification of a few details needed. Specific comments follow.
- More description is required how the proposed protocol is secure in the IND-PrID-CPA notion.
Revisions with respect to the above comments:
Thank you for the comment. We have explained how the proposed scheme is secure in the IND-PrID-CPA notion in Section 5 as follows:
“The concept of this security proof is a technique of proof by contradiction. That is, we first assume that there is an adversary who is able to break the proposed scheme under the adaptively chosen identity and chosen plaintext attacks. Then we could take the advantage of this adversary to break a well-known intractable cryptographic assumption with non-negligible advantage. Since there is no efficient polynomial-time algorithm that could solve any well-known cryptographic assumption, we conclude that our initial assumption is wrong, which also completes the whole security proof.”
- More precise formulation needed how proposed mechanism is able to withstand the adaptive adversary in the random oracle models.
Revisions with respect to the above comments:
Thank you for the comment. In Section 5, we have stated that the chance for B to solve the problem of DBDH could be written as e × Pr[Good] and added some precise formulations as follows:
Pr[Good] = Pr[ØKeyExtract] × Pr[ØRKgen] × Pr[ØChallenge]
£ (t)qke(t)qrk(1 - t)
= (t)qke + qrk(1 - t).
=
Consequently, the constructed algorithm B has a non-negligible advantage e' ³ to break the DBDH problem, which is a contradiction to our initial assumption. In other words, the proposed mechanism is able to withstand the adaptive adversary in the random oracle models.
- In point 5. Efficiency and Comparison The detailed of the simulation environment is required.
Revisions with respect to the above comments:
Thank you for the comment. We have added a new table (Table 1) to show the simulation environments as follows:
Table 1. Simulation environments
|
Item |
Environment |
|
Processor |
Intel Core 2 Duo @ 2.1Ghz |
|
Memory size |
2GB |
|
Operating system |
Linux Ubuntu version 9.1 |
|
Software |
PBC library [22] |
Expanded References
- Chen, Z. Research on Internet security situation awareness prediction technology based on improved RBF neural network algorithm, Journal of Computational and Cognitive Engineering 2022, 1, 103-108.
- Gutub, A.; Gong, M. Boosting image watermarking authenticity spreading secrecy from counting-based secret-sharing. CAAI Transactions on Intelligence Technology 2022, 1-13. https://doi.org/10.1049/cit2.12093
- Pavithran, P.; Mathew, S.; Namasudra, S.; Srivastava, G. A novel cryptosystem based on DNA cryptography, hyperchaotic systems and a randomly generated Moore machine for cyber physical systems, Computer Communications 2022, 188, 1-12.
- Mahmood, Z. H.; Ibrahem, M.K. New fully homomorphic encryption scheme based on multistage partial homomorphic encryption applied in cloud computing, In Proceedings of 2018 1st Annual International Conference on Information and Sciences (AiCIS), 2018, pp. 182-186.
- Dostalek, L.; Safarik, J. Strong password authentication with AKA authentication mechanism, In Proceedings of 2017 International Conference on Applied Electronics (AE), 2017, pp. 1-6.
- Sarkar, M.; Saha, K.; Namasudra, S.; Roy, P. An efficient and time saving web service based android application, SSRG International Journal of Computer Science and Engineering 2015, 2, 18-21.
- Kumari, S.; Kumar, R.; Kadry, S.; Namasudra, S.; Taniar, D. Maintainable stochastic communication network reliability within tolerable packet error rate, Computer Communications 2021, 178, 161-168.
- Wani, A.; Revathi, S.; Khaliq, R. SDN-based intrusion detection system for IoT using deep learning classifier (IDSIoT-SDL), CAAI Transactions on Intelligence Technology 2021, 6, 281-290.
- Bajaj, K.; Sharma, B.; Singh, R. Comparative analysis of simulators for IoT applications in fog/cloud computing, In Proceedings of the 2022 International Conference on Sustainable Computing and Data Communication Systems (ICSCDS), 2022, pp. 983-988.
- Tseng, C.L.; Lin, F.J. Extending scalability of IoT/M2M platforms with fog computing, In Proceedings of 2018 IEEE 4th World Forum on Internet of Things (WF-IoT), 2018, pp. 825-830.
- Verma, R.; Kumari, A.; Anand, A.; Yadavalli, V.S.S. Revisiting shift cipher technique for amplified data security, Journal of Computational and Cognitive Engineering 2022, 00, 1-7.

Reviewer 2 Report
There are some problems need to be improved.
--The introduction section introduces too much common sense and suggests modification.
--The related works section does not summarize and refine the literature. It is also necessary for the author to re-summarize which work is the basis of this study, which can be compared in this article, and which is the sublimation of this study.
--Punctuation in Equation (11) is suggested to be deleted.
-- Experimental analysis is too simplistic.
-- There are a small number of spelling errors and grammatical errors.
Author Response
The authors would like to thank reviewers for their valuable suggestions that result in the improvement of correctness and readability of this manuscript. Revisions with respect to each comment are listed below:
Comments of Reviewer 2:
There are some problems need to be improved.
- The introduction section introduces too much common sense and suggests modification.
Revisions with respect to the above comments:
Thank you for the comment. We have followed the Security-Guidance-v4.0 and the NIST SP 800-145 to clarify cloud deployment models and improve the description of service models in the introductory section. Please refer to our revision for more details.
- The related works section does not summarize and refine the literature. It is also necessary for the author to re-summarize which work is the basis of this study, which can be compared in this article, and which is the sublimation of this study.
Revisions with respect to the above comments:
Thank you for the comment. We have clarified these issues in the section of related works as follows:
“In 2022, Lin et al. [25] improved Zhang et al.’s scheme [23] by eliminating some security flaws. They also showed that their enhanced scheme maintains the properties to revoke invalid users and generate private keys anonymously. Although their work has improved security which is provably secure in the random oracle model, the computational complexity is still high. Motivated by this challenging problem, we will propose a more efficient PRE scheme based on the study of Lin et al. [25]. Up to the present, there have been several PRE mechanisms [26-32] proposed for different applications. However, only a few works [19, 23, 25] take the issue of cloud computing or fog computing scenarios into consideration. We will compare the proposed mechanism with these schemes and show the computational advantage of ours in the later section.”
- Punctuation in Equation (11) is suggested to be deleted.
Revisions with respect to the above comments:
Thank you for the comment. We have deleted the punctuation in Equation (11).
- Experimental analysis is too simplistic.
Revisions with respect to the above comments:
Thank you for the comment. In general, the experimental analysis of a cryptographic protocol is evaluated in terms of the time complexity. Hence, we compare the proposed mechanism with several related works in Table 4 and Fig. 2. For better understanding of our simulation environments, we have added a new table (Table 2) to show the details of simulation environments as follows:
Table 2. Simulation environments
|
Item |
Environment |
|
Processor |
Intel Core 2 Duo @ 2.1Ghz |
|
Memory size |
2GB |
|
Operating system |
Linux Ubuntu version 9.1 |
|
Software |
PBC library [33] |
- There are a small number of spelling errors and grammatical errors.
Revisions with respect to the above comments:
Thank you for the comment. We have corrected the spelling errors and grammatical errors throughout the revision.
Expanded References
- Chen, Z. Research on Internet security situation awareness prediction technology based on improved RBF neural network algorithm, Journal of Computational and Cognitive Engineering 2022, 1, 103-108.
- Gutub, A.; Gong, M. Boosting image watermarking authenticity spreading secrecy from counting-based secret-sharing. CAAI Transactions on Intelligence Technology 2022, 1-13. https://doi.org/10.1049/cit2.12093
- Pavithran, P.; Mathew, S.; Namasudra, S.; Srivastava, G. A novel cryptosystem based on DNA cryptography, hyperchaotic systems and a randomly generated Moore machine for cyber physical systems, Computer Communications 2022, 188, 1-12.
- Mahmood, Z. H.; Ibrahem, M.K. New fully homomorphic encryption scheme based on multistage partial homomorphic encryption applied in cloud computing, In Proceedings of 2018 1st Annual International Conference on Information and Sciences (AiCIS), 2018, pp. 182-186.
- Dostalek, L.; Safarik, J. Strong password authentication with AKA authentication mechanism, In Proceedings of 2017 International Conference on Applied Electronics (AE), 2017, pp. 1-6.
- Sarkar, M.; Saha, K.; Namasudra, S.; Roy, P. An efficient and time saving web service based android application, SSRG International Journal of Computer Science and Engineering 2015, 2, 18-21.
- Kumari, S.; Kumar, R.; Kadry, S.; Namasudra, S.; Taniar, D. Maintainable stochastic communication network reliability within tolerable packet error rate, Computer Communications 2021, 178, 161-168.
- Wani, A.; Revathi, S.; Khaliq, R. SDN-based intrusion detection system for IoT using deep learning classifier (IDSIoT-SDL), CAAI Transactions on Intelligence Technology 2021, 6, 281-290.
- Bajaj, K.; Sharma, B.; Singh, R. Comparative analysis of simulators for IoT applications in fog/cloud computing, In Proceedings of the 2022 International Conference on Sustainable Computing and Data Communication Systems (ICSCDS), 2022, pp. 983-988.
- Tseng, C.L.; Lin, F.J. Extending scalability of IoT/M2M platforms with fog computing, In Proceedings of 2018 IEEE 4th World Forum on Internet of Things (WF-IoT), 2018, pp. 825-830.
- Verma, R.; Kumari, A.; Anand, A.; Yadavalli, V.S.S. Revisiting shift cipher technique for amplified data security, Journal of Computational and Cognitive Engineering 2022, 00, 1-7.

Reviewer 3 Report
Comments and Suggestions for Authors
1. Regarding the classification of cloud service according to the location of stored data (line 32+), the authors should clarify that Cloud is not only for storage and the deployment models influences the managing, owning, location and users of the cloud (see Security-Guidance-v4.0 https://cloudsecurityalliance.org/education/ccsk/study-guide/ page 12).
2. The description of the service models (line 44+) can be improved (see SP 800-145 The NIST Definition of Cloud Computing https://nvlpubs.nist.gov/nistpubs/Legacy/SP/nistspecialpublication800-145.pdf)
3. The Effeciency and Comparison was performed with respect to Han et al. [19], the work of Han et al. is not described in the Related Works section
Author Response
The authors would like to thank reviewers for their valuable suggestions that result in the improvement of correctness and readability of this manuscript. Revisions with respect to each comment are listed below:
Comments of Reviewer 3:
Comments and Suggestions for Authors
- Regarding the classification of cloud service according to the location of stored data (line 32+), the authors should clarify that Cloud is not only for storage and the deployment models influences the managing, owning, location and users of the cloud (see Security-Guidance-v4.0 https://cloudsecurityalliance.org/education/ccsk/study-guide/ page 12).
Revisions with respect to the above comments:
Thank you for the comment. In the introductory section, we have added the following statements to clarify the cloud deployment models as follows:
“Clouds are not only for storage and the deployment model influences the managing, owning, location and users of the cloud. According to the location of stored data and how the technologies are deployed and consumed, we could classify cloud service models into three different kinds as follows.”
- The description of the service models (line 44+) can be improved (see SP 800-145 The NIST Definition of Cloud Computing https://nvlpubs.nist.gov/nistpubs/Legacy/SP/nistspecialpublication800-145.pdf)
Revisions with respect to the above comments:
Thank you for the comment. We have improved the description of service models according to the NIST SP 800-145 as follows:
- Software as a Service (SaaS): It is the most common model in which users can utilize all kinds of interfaces (including web-based or program-based) to acquire the resources and web services [7] such as stream media platforms running on a cloud infrastructure. The advantage of this model is that users do not have to be responsible for controlling or maintaining the cloud infrastructure like communication networks [8], operating systems, storage and applications.
- Platform as a Service (PaaS): In this model, the cloud service provider is responsible for providing application development platforms such as storage capacity, computing resources, programming languages, libraries and related development tools, etc. Users can utilize these tools to deploy consumer-created application programs on the cloud infrastructure and they do not have to control or maintain the cloud infrastructure.
- Infrastructure as a Service (IaaS): The cloud service provider will supply users with all kinds of storage, computing and network resources and users could utilize these infrastructures to deploy their own platforms and application programs. The advantage of this model is that users do not have to control or maintain the cloud infrastructure, but have the control over their deployed applications, storage and operating systems.
- The Efficiency and Comparison was performed with respect to Han et al. [19], the work of Han et al. is not described in the Related Works section.
Revisions with respect to the above comments:
Thank you for the comment. In the Section of related works, we have mentioned the work of Han et al. as follows:
“…Considering the queries from intra-domain and inter-domain in the cloud computing scenario, Han et al. [19] proposed an identity-based PRE scheme. Their scheme is secure against collusion attacks and the access permission could be made by the data owner, rather than by the central authority. However, the computational complexity of their scheme is high.”
Expanded References
- Chen, Z. Research on Internet security situation awareness prediction technology based on improved RBF neural network algorithm, Journal of Computational and Cognitive Engineering 2022, 1, 103-108.
- Gutub, A.; Gong, M. Boosting image watermarking authenticity spreading secrecy from counting-based secret-sharing. CAAI Transactions on Intelligence Technology 2022, 1-13. https://doi.org/10.1049/cit2.12093
- Pavithran, P.; Mathew, S.; Namasudra, S.; Srivastava, G. A novel cryptosystem based on DNA cryptography, hyperchaotic systems and a randomly generated Moore machine for cyber physical systems, Computer Communications 2022, 188, 1-12.
- Mahmood, Z. H.; Ibrahem, M.K. New fully homomorphic encryption scheme based on multistage partial homomorphic encryption applied in cloud computing, In Proceedings of 2018 1st Annual International Conference on Information and Sciences (AiCIS), 2018, pp. 182-186.
- Dostalek, L.; Safarik, J. Strong password authentication with AKA authentication mechanism, In Proceedings of 2017 International Conference on Applied Electronics (AE), 2017, pp. 1-6.
- Sarkar, M.; Saha, K.; Namasudra, S.; Roy, P. An efficient and time saving web service based android application, SSRG International Journal of Computer Science and Engineering 2015, 2, 18-21.
- Kumari, S.; Kumar, R.; Kadry, S.; Namasudra, S.; Taniar, D. Maintainable stochastic communication network reliability within tolerable packet error rate, Computer Communications 2021, 178, 161-168.
- Wani, A.; Revathi, S.; Khaliq, R. SDN-based intrusion detection system for IoT using deep learning classifier (IDSIoT-SDL), CAAI Transactions on Intelligence Technology 2021, 6, 281-290.
- Bajaj, K.; Sharma, B.; Singh, R. Comparative analysis of simulators for IoT applications in fog/cloud computing, In Proceedings of the 2022 International Conference on Sustainable Computing and Data Communication Systems (ICSCDS), 2022, pp. 983-988.
- Tseng, C.L.; Lin, F.J. Extending scalability of IoT/M2M platforms with fog computing, In Proceedings of 2018 IEEE 4th World Forum on Internet of Things (WF-IoT), 2018, pp. 825-830.
- Verma, R.; Kumari, A.; Anand, A.; Yadavalli, V.S.S. Revisiting shift cipher technique for amplified data security, Journal of Computational and Cognitive Engineering 2022, 00, 1-7.

Reviewer 4 Report
1. Motivations of the paper are not clear.
2. Contributions are not mentioned. Most importantly, the structure of the Introduction section is very poor.
3. Related schemes are not discussed properly. They must be represented in a different section. Also, include the following references:
a) SDN-based intrusion detection system for IoT using deep learning classifier (IDSIoT-SDL)
b) Revisiting shift cipher technique for amplified data security
c) An efficient and time saving web service based android application
d) Boosting image watermarking authenticity spreading secrecy from counting-based secret-sharing
e) Maintainable stochastic communication network reliability within tolerable packet error rate
f) Research on internet security situation awareness prediction technology based on improved RBF neural network algorithm
g) A novel cryptosystem based on DNA cryptography, hyperchaotic systems and a randomly generated Moore machine for cyber physical systems
4. The proposed algorithm is not properly represented.
5. The proposed scheme is unstructured. It is hard to identify the novelty of the proposed work.
6. Equations and figures are not represented properly. The key terms of all the equations must be defined.
7. Technical discussion on results is not given.
8. Moreover, the results are not convincing.
9. The English language is very poor.
10. The organization of the paper is poor.
11. Important references are missing and all the details of the references are not given.
Author Response
The authors would like to thank reviewers for their valuable suggestions that result in the improvement of correctness and readability of this manuscript. Revisions with respect to each comment are listed below:
Comments of Reviewer 4:
- Motivations of the paper are not clear.
Revisions with respect to the above comments:
Thank you for the comment. In the section of related works, we have added the motivations of this paper as follows:
“In 2022, Lin et al. [25] improved Zhang et al.’s scheme [23] by eliminating some security flaws. They also showed that their enhanced scheme maintains the properties to revoke invalid users and generate private keys anonymously. Although their work has improved security which is provably secure in the random oracle model, the computational complexity is still high. Motivated by this challenging problem, we will propose a more efficient PRE scheme based on the study of Lin et al. [25]. Up to the present, there have been several PRE mechanisms [26-32] proposed for different applications. However, only a few works [19, 23, 25] take the issue of cloud computing or fog computing scenarios into consideration. We will compare the proposed mechanism with these schemes and show the computational advantage of ours in the later section.”
- Contributions are not mentioned. Most importantly, the structure of the Introduction section is very poor.
Revisions with respect to the above comments:
Thank you for the comment. We have described our contributions at the end of the Section of related works as follows:
“The main contribution of this study is to propose an identity-based PRE scheme for the fog computing scenario and using the technique of anonymous key generation. In the proposed system, we use public channels for key distribution and avoid the troublesome problem of key escrow. In addition, the decision of access privilege for cloud ciphertexts is controlled by the data owner, rather than by the central authority. Moreover, we will demonstrate that the proposed protocol is not only IND-PrID-CPA secure, but also has lower computational costs.”
Also, in the introductory section, we would like to describe why the cloud/fog computing scenario is important for the proposed mechanism. Therefore, we have followed the Security-Guidance-v4.0 and the NIST SP 800-145 to clarify cloud deployment models and improve the description of service models in the introductory section. Please refer to our revision for more details.
- Related schemes are not discussed properly. They must be represented in a different section. Also, include the following references:
- a) SDN-based intrusion detection system for IoT using deep learning classifier (IDSIoT-SDL)
- b) Revisiting shift cipher technique for amplified data security
- c) An efficient and time saving web service based android application
- d) Boosting image watermarking authenticity spreading secrecy from counting-based secret-sharing
- e) Maintainable stochastic communication network reliability within tolerable packet error rate
- f) Research on internet security situation awareness prediction technology based on improved RBF neural network algorithm
- g) A novel cryptosystem based on DNA cryptography, hyperchaotic systems and a randomly generated Moore machine for cyber physical systems
Revisions with respect to the above comments:
Thank you for the comment. We have presented the related works in a separate section. Moreover, the suggested references are also cited in the revision.
- The proposed algorithm is not properly represented.
Revisions with respect to the above comments:
Thank you for the comment. We have re-checked the correctness of all equations and modified unclear symbols.
- The proposed scheme is unstructured. It is hard to identify the novelty of the proposed work.
Revisions with respect to the above comments:
Thank you for the comment. We have highlighted the novelty of the proposed work in our contributions which are described at the end of the Section of related works as follows:
“The main contribution of this study is to propose an identity-based PRE scheme for the fog computing scenario and using the technique of anonymous key generation. In the proposed system, we use public channels for key distribution and avoid the troublesome problem of key escrow. In addition, the decision of access privilege for cloud ciphertexts is controlled by the data owner, rather than by the central authority. Moreover, we will demonstrate that the proposed protocol is not only IND-PrID-CPA secure, but also has lower computational costs.”
- Equations and figures are not represented properly. The key terms of all the equations must be defined.
Revisions with respect to the above comments:
Thank you for the comment. In Section 4.3, we have added a table of symbol notations to define the key terms of all equations as follows:
Table 1. Symbol notations
|
Notation |
Description |
|
l |
security value |
|
G1, G2 |
groups of prime order p |
|
g |
a generator of G1 |
|
e |
a bilinear map satisfying e: G1 ´ G1 ® G2 |
|
s |
master secret key |
|
Q |
master public key satisfying Q = gs |
|
RL |
revocation list |
|
h1, h2, h3 |
collision-resistant hash functions |
|
F |
system public information |
|
SK |
symmetric key |
|
E(×)/D(×) |
symmetric encryption/decryption function |
|
{r1, r2, r3} |
ciphertext |
|
Cind |
data category name |
|
(w1, w2, w3) |
re-encryption key |
|
(r1', r2, r3, r4', r5') |
re-encrypted ciphertext |
- Technical discussion on results is not given.
Revisions with respect to the above comments:
Thank you for the comment. The technical discussion of a cryptographic protocol includes algorithm presentation, security proof and efficiency analysis. We have addressed these issues in separate sections. Please refer to our revision for more details.
- Moreover, the results are not convincing.
Revisions with respect to the above comments:
Thank you for the comment. In this revision, we have modified unclear algorithms and the section of security proof to make the results of this research more convincing.
- The English language is very poor.
Revisions with respect to the above comments:
Thank you for the comment. We have corrected the spelling errors and grammatical errors throughout the revision.
- The organization of the paper is poor.
Revisions with respect to the above comments:
Thank you for the comment. We have modified the organization of our revision by adding an independent section of related works.
- Important references are missing and all the details of the references are not given.
Revisions with respect to the above comments:
Thank you for the comment. We have added some suggested references that are relevant to the researched topic in the revision.
Expanded References
- Chen, Z. Research on Internet security situation awareness prediction technology based on improved RBF neural network algorithm, Journal of Computational and Cognitive Engineering 2022, 1, 103-108.
- Gutub, A.; Gong, M. Boosting image watermarking authenticity spreading secrecy from counting-based secret-sharing. CAAI Transactions on Intelligence Technology 2022, 1-13. https://doi.org/10.1049/cit2.12093
- Pavithran, P.; Mathew, S.; Namasudra, S.; Srivastava, G. A novel cryptosystem based on DNA cryptography, hyperchaotic systems and a randomly generated Moore machine for cyber physical systems, Computer Communications 2022, 188, 1-12.
- Mahmood, Z. H.; Ibrahem, M.K. New fully homomorphic encryption scheme based on multistage partial homomorphic encryption applied in cloud computing, In Proceedings of 2018 1st Annual International Conference on Information and Sciences (AiCIS), 2018, pp. 182-186.
- Dostalek, L.; Safarik, J. Strong password authentication with AKA authentication mechanism, In Proceedings of 2017 International Conference on Applied Electronics (AE), 2017, pp. 1-6.
- Sarkar, M.; Saha, K.; Namasudra, S.; Roy, P. An efficient and time saving web service based android application, SSRG International Journal of Computer Science and Engineering 2015, 2, 18-21.
- Kumari, S.; Kumar, R.; Kadry, S.; Namasudra, S.; Taniar, D. Maintainable stochastic communication network reliability within tolerable packet error rate, Computer Communications 2021, 178, 161-168.
- Wani, A.; Revathi, S.; Khaliq, R. SDN-based intrusion detection system for IoT using deep learning classifier (IDSIoT-SDL), CAAI Transactions on Intelligence Technology 2021, 6, 281-290.
- Bajaj, K.; Sharma, B.; Singh, R. Comparative analysis of simulators for IoT applications in fog/cloud computing, In Proceedings of the 2022 International Conference on Sustainable Computing and Data Communication Systems (ICSCDS), 2022, pp. 983-988.
- Tseng, C.L.; Lin, F.J. Extending scalability of IoT/M2M platforms with fog computing, In Proceedings of 2018 IEEE 4th World Forum on Internet of Things (WF-IoT), 2018, pp. 825-830.
- Verma, R.; Kumari, A.; Anand, A.; Yadavalli, V.S.S. Revisiting shift cipher technique for amplified data security, Journal of Computational and Cognitive Engineering 2022, 00, 1-7.

Round 2
Reviewer 2 Report
The author has revised the relevant questions.
Reviewer 4 Report
All the previous comments are addressed.